# Juvenile Idiopathic Arthritis, Uveitis and Multiple Sclerosis: Description of Two Patients and Literature Review

**DOI:** 10.3390/biomedicines10082041

**Published:** 2022-08-21

**Authors:** Cecilia Beatrice Chighizola, Matteo Ferrito, Luca Marelli, Irene Pontikaki, Paolo Nucci, Elisabetta Miserocchi, Roberto Caporali

**Affiliations:** 1Department of Clinical Sciences and Community Health, University of Milan, 20122 Milan, Italy; 2Paediatric Rheumatology Unit, ASST G. Pini & CTO, 20122 Milan, Italy; 3Clinical Rheumatology Unit, ASST G. Pini & CTO, 20122 Milan, Italy; 4Eye Clinic, San Giuseppe Hospital, IRCCS Multimedica, 20123 Milan, Italy; 5Department of Biomedical, Surgical and Dental Sciences, University of Milan, 20122 Milan, Italy; 6Uveitis Service, IRCCS Ospedale San Raffaele, Università Vita-Salute, Via Olgettina 60, 20132 Milan, Italy

**Keywords:** juvenile idiopathic arthritis, multiple sclerosis, anti-TNF-α agents, comorbidity, treatment

## Abstract

Juvenile idiopathic arthritis (JIA) is the most common rheumatic disease in childhood, while multiple sclerosis (MS) is a demyelinating disease of the central nervous system, characterized by remission and exacerbation phases. An association between MS and rheumatologic diseases, in particular rheumatoid arthritis, has been described and numerous studies acknowledge anti-TNF-α drugs as MS triggers. Conversely, the association between MS and JIA has been reported merely in five cases in the literature. We describe two cases of adult patients with longstanding JIA and JIA-associated uveitis, who developed MS. The first patient was on methotrexate and adalimumab when she developed dizziness and nausea. Characteristic MRI lesions and oligoclonal bands in cerebrospinal fluid led to MS diagnosis. Adalimumab was discontinued, and she was treated with three pulses of intravenous methylprednisolone. After a few months, rituximab was started. The second patient had been treated with anti-TNF-α and then switched to abatacept. She complained of unilateral arm and facial paraesthesias; brain MRI showed characteristic lesions, and MS was diagnosed. Three pulses of intravenous methylprednisolone were administered; neurological disease remained stable, and abatacept was reintroduced. Further studies are warranted to define if there is an association between JIA and MS, if MS represents JIA comorbidity or if anti-TNF-α underpins MS development.

## 1. Introduction

Comorbidities are defined as distinct additional diseases that exist prior to or during the clinical course of a given index condition, with some being transient and others remaining active and persistent. Comorbidities can manifest before, after or concurrently with the index disease [1]. In the context of systemic autoimmune conditions, the coexistence of two or more diseases in a single patient occurs relatively frequently. The two coexisting conditions may be simply coincidental or share the same predisposing background. In adults with rheumatoid arthritis (RA), comorbidities are common, and approximately 75% of patients carry a second comorbid diagnosis [2]. Among patients with juvenile idiopathic arthritis (JIA), the most common rheumatic disease presenting in childhood, the impact of comorbidities has yet to be characterized. 

It would be extremely important to clarify the exact comorbidity burden in JIA, as all these concomitant diseases might contribute to the overall impact for patients in terms of socio-economic, cultural, environmental, and psychological implications. First, studies should correctly identify a “comorbidity”. Comorbidities do not consist of consequences of the index disease: it is thus incorrect to regard uveitis or macrophage activation syndrome as comorbidities in JIA, but these should be rather enlisted as extra-articular manifestations of JIA [3,4]. In addition, comorbidities should be differentiated from complications of an index disease, which consists in adverse events occurring after diagnosis of that condition. This is the case of infections, which are adverse events that most commonly have to be ascribed to treatment of underlying JIA [4].

When back in the early 2000s, biological agents have been marketed for JIA treatment; the scenario has become even more complex, given the pleiotropic effects of these pharmacological tools and the plethora of conditions that might supervene in treated patients. Important data can be extracted from collaborative registries, but even everyday clinical practice might provide crucial hints into potentially relevant comorbidities and treatment-related side effects. 

In the present manuscript, we describe two cases of adult patients with longstanding JIA complicated by uveitis, who developed neurological deficits with evidence of demyelinating lesions at imaging, diagnosed as multiple sclerosis (MS). Both patients were followed at our institution in a dedicated rheumatology transition clinic, an outpatient third-level referral centre where approximately 300 patients with JIA are followed once reaching adulthood. In RA patients, the interplay with MS has been progressively elucidated, as we will discuss in detail in the text. MS is not currently regarded as a comorbid condition for RA, since it is now clear that such a relationship is mediated by the exposure to agents targeting anti-tumour necrosis factor a (TNF-α). Much less evidence is available in the setting of JIA; thus, we believe it is timely to conduct an extensive literature review about the potential relationships between JIA, uveitis, MS and TNF-α inhibitors. 

### 1.1. Juvenile Idiopathic Arthritis: An Overview

JIA is defined by the presence of arthritis of unknown aetiology lasting for more than 6 weeks and starting before 16 years of age [5]. The estimated prevalence of JIA ranges from 3.8 to 400 cases per 100,000 people [6]. JIA includes a group of complex and heterogeneous diseases: currently, the International League Against Rheumatic Diseases (ILAR) classification criteria acknowledge seven JIA subtypes, based upon predominant clinical and laboratory features and the number of involved joints at disease onset: oligoarticular JIA, rheumatoid factor (RF)-negative polyarticular JIA, RF-positive polyarticular JIA, enthesitis-related arthritis (ERA), psoriatic arthritis, systemic JIA and undifferentiated arthritis [7].

Oligoarticular JIA represents the most common form (>50% of patients). It is defined by the involvement of no more than four joints during the first 6 months from symptom onset. Patients with oligoarticular involvement are subclassified according to anti-nuclear antibodies (ANA) positivity. After 6 months from disease onset, patients with oligoarticular JIA can have a disease still confined to four joints (persistent oligoarticular JIA) or progress to a polyarticular involvement (extended oligoarticular JIA) [8].

Polyarthritis is defined as arthritis occurring in more than four joints. Two types of polyarticular JIA are recognized upon RF positivity. RF-positive polyarticular JIA is rare (5% of patients with JIA) and substantially mirrors adult-onset seropositive RA, while RF-negative polyarticular JIA is more common (20% of patients) and often originates from extended oligoarthritis [9,10,11].

Psoriatic JIA and ERA together account for 10–20% of cases and share a similar pattern of joint distribution, affecting most commonly lower limb joints with an asymmetrical pattern. Furthermore, two distinctive features of these subsets are axial skeleton involvement (sacroiliitis) and enthesitis, which is inflammation of tendons, ligaments, fasciae, and joint capsule insertion on the periosteum, most frequently plantar fascia, Achilleon tendon, and tibial insertion of the patellar tendon. In psoriatic arthritis, as for the adult form, the diagnosis requires the concomitance of psoriatic skin lesions or nail involvement, history of at least one of these features, or in the case of psoriasis, affecting a first-degree relative [10,12].

Systemic JIA represents 10% of all JIA cases, and its clinical presentation is mediated by a sustained systemic inflammatory response: high spike periodic fever, skin salmon-coloured rash, arthritis and neutrophilic leucocytosis. Typically, longstanding systemic JIA resembles polyarticular JIA, since systemic symptoms tend to recede over time while the articular disease is more often persistent [12]. However, the impact of JIA goes well beyond the joints even in subsets other than the systemic form: the so-called extra-articular manifestations of JIA include uveitis, cardiovascular morbidity, pulmonary involvement, neurological lesions and gastrointestinal symptoms [9].

Notwithstanding the considerable recent progress in unravelling the mechanisms for local and systemic inflammation in JIA, the initial triggers responsible for the whole process are still largely unknown, even though the early onset of rheumatic diseases in children suggests a large contribution of genetic factors [9]. Several cell types are involved in joint damage, such as monocytes/macrophages, granulocytes, B and T lymphocytes, synovial fibroblasts, and osteoclasts. In particular, in JIA, different subsets of CD4+ T cells play a pivotal role, and the synovial fluid from an affected joint contains classic T helper (Th) 1, non-classic Th1 and Th17 cells. Different cytokines produced by the above-cited cell types drive the inflammatory process: TNF-α, interleukin (IL)-1, IL-6, and IL-17 [9]. To note, systemic JIA differs from the other subsets since it acknowledges as the predominant pathogenic driver a dysregulation of the innate immune response. Thus, systemic JIA is considered a hybrid entity due to its autoinflammatory features [12,13]. 

The goal of treatment in JIA consists in the prevention of joint damage by obtaining a full remission of disease activity, namely the absence of clinical signs and symptoms and normal inflammatory markers. To achieve clinical remission, long-term drug treatment is usually required, including systemic and intraarticular corticosteroids, disease-modifying anti-rheumatic drugs (DMARDs, e.g., methotrexate) and, when these are not sufficient, novel biologic or small drugs which target different cytokines and cellular structures [9]. 

The introduction of biological and small drugs has revolutionized the therapeutic approach to JIA, allowing for immense progresses in the management of patients unresponsive to first-line therapies [14]. Despite the revolutionized therapeutic approach, in at least half of cases, childhood-onset arthritis can persist into adulthood [15]. Thus, patients with persisting JIA are exposed to long-lasting treatment with synthetic and biologic DMARDs, whose impact in the long term still needs to be further clarified.

### 1.2. Beyond the Joints: Juvenile Idiopathic Arthritis and Uveitis 

Uveitis consists in the inflammation of the uvea, which is the middle, pigmented, vascular layer of the eye including the iris, ciliary body, and choroid (Figure 1). 

It is the most common extra-articular manifestation of JIA [16]. According to the Standardization of Uveitis Nomenclature Workshop (SUN) criteria, uveitis can be classified based on the primary site of inflammation, onset, duration, and course of the disease. Anatomically, uveitis can be described as anterior, intermediate, posterior or panuveitis. The temporal pattern is classified as acute, subacute, chronic, or recurrent [17]. The most common form of intraocular inflammation associated with JIA is chronic anterior uveitis, which can be unilateral or bilateral. Acute anterior uveitis can also occur in JIA, and is usually associated with ERA and HLA-B27 positivity [18]. The reported prevalence of uveitis among children with JIA varies from 11.6% to 30%. Several risk factors for JIA have been identified: ANA positivity, younger age (<6 years of age) at the onset of arthritis, oligoarticular subset, and female gender are the main ones [18]. Importantly, JIA-associated uveitis could be often asymptomatic and, even in case of symptoms, children could experience difficulties in reporting their complains. Thus, a strict ophthalmologic screening is warranted every 3 to 12 months, with a tailored regimen according to each patient’s risk profile [19]. Although frequently asymptomatic at onset, JIA-associated uveitis may lead to severe vision-threatening complications such as band keratopathy, cataract, glaucomatous optic neuropathy and macular oedema, potentially impacting on patient ability to perform tasks that rely on vision [20]. 

The exact aetiology and pathophysiology of JIA-associated uveitis remain hypothetical, as well as the relationship between the ocular and articular inflammation. Nevertheless, a genetic predisposition, particularly with respect to HLA class II genes, has been proposed to play a role, despite the low concordance rate among twins [21]. Non-infectious uveitis is regarded as a T-cell-mediated disease involving CD4+ Th1 and Th17 subsets; however, no evidence of direct T-cell involvement has yet been raised in JIA-associated uveitis [21]. A recent immunohistochemical study conducted on iridectomy specimens from patients with JIA uveitis revealed plasma cells to be abundant, while CD4+ and CD8+ cells were not always detectable, even if a modest predominance of the former was observed [22]. Conversely, the levels in the aqueous humour of several proinflammatory cytokines and chemokines as IL-2, IL-6, IL-13, IL-18, IFN-γ, TNF, soluble ICAM-1 (also known as CD54), C-C motif chemokine 5 (CCL5, also known as RANTES) and C-X-C motif chemokine 10 (CXCL10, also known as IP-10) were considerably higher in 11 children with JIA-associated uveitis compared to controls without uveitis [23].

### 1.3. Multiple Sclerosis: An Overview

MS is a chronic neurodegenerative disease of the central nervous system (CNS) characterised by inflammatory and demyelinating processes leading to axonal degeneration [24]. It affects approximately 2.5 million people globally, with a higher prevalence in western countries (1 per 1000 individuals). The female-to-male ratio is 3:1 [25]; disease onset usually occurs between 20 and 40 years of age, although in up to 10% of cases, disease manifests during adolescence [26]. Both genetic and environmental risk factors contribute to the pathogenesis of MS. Several polymorphisms involved in the activation of the innate and adaptive immune responses have been shown to confer a higher risk of developing MS, even if the environment seems to weigh more than the genetic background [27]. The most important environmental risk factors for MS are Epstein–Barr virus (EBV) infection, tobacco consumption or passive smoking, hypovitaminosis D and obesity [28]. A large longitudinal study comprising more than 10 million young United States military has recently identified EBV infection as a strong and independent risk factor for MS, whereas no other virus conferred an increased hazard. In particular, EBV infection raised the risk of MS by 32-fold [29]. It has been proposed that EBV might augment the risk of MS by driving B lymphocyte activation and proliferation, although this topic remains controversial [30]. Fat-soluble vitamin D (ergocalciferol-D2, cholecalciferol-D3, alfacalcidol) exerts an immunomodulatory effect by inhibiting plasma cell differentiation and immunoglobulin production in vitro [31]. The evidence of vitamin D deficiency as a potential risk factor for MS emerged largely from observational studies, although such association is supported even by a post hoc analysis of two phase 3 trials [32]. Furthermore, vitamin D deficiency seems to be a risk factor for disease onset as well as progression; a longitudinal study reported that each 10 ng/mL increase in vitamin D serum level reduces the risk of developing new demyelinating lesions by nearly 15% [33]. Lastly, smokers carry a higher risk of developing MS than non-smokers (relative risk [RR] 1.5). Similarly to other diseases, cigarette smoking increases the hazard of not only MS development but also the rate of disease progression [34,35]. In MS, uncontrolled inflammation and autoimmunity against central myelin is driven by blood-borne autoreactive T lymphocytes. These cells are stimulated by antigen-presenting cells with putative exogenous antigens and are equipped, due to molecular mimicry, against epitopes of basic myelin protein expressed by CNS oligodendrocytes. Such autoreactive T cells are found infiltrating in the plaques, releasing several pro-inflammatory cytokines such as TNF-α, which in turn activate resident microglia leading to further release of proinflammatory cytokines. Microglia, endothelial cells, and astrocytes from patients with MS have all been found to upregulate TNF [36,37]. The meningeal inflammation characteristic of MS was found to be associated with a shift in the balance of TNF signalling away from NFkB-mediated anti-apoptotic pathways towards RIPK3-mediated pro-apoptotic/pro-necroptotic signalling in the grey matter [38]. Cutting-edge research has clarified the association between exonic variants of the TNFRSF1A gene, which encodes one of the TNF-α receptors, in particular the variant rs4149584, with the risk of developing MS [39]. In agreement with this burden of data, TNF levels in the cerebrospinal fluid has been recently proposed as a biomarker of disease activity and as a tool to predict response to treatment [40].

The diagnosis of MS can be formulated in patients with symptoms suggestive of MS, once alternative diagnoses have been ruled out, based upon the radiological demonstrations of lesions in the CNS that are disseminated both in space and in time, according to the updated McDonald’s criteria (Table 1). MS demyelinating attacks can involve optic nerves, periventricular white matter, brainstem, cerebellum, and spinal cord [24]. 

The most common manifestations in patients with MS are represented by sensory, visual, motor, brainstem and cerebellar deficits and sexual or sphincter disturbances (Table 2). Sensory symptoms consist mainly in sensory loss (nearly 67%) and paraesthesias (nearly 33%) usually involving the limbs; Lhermitte’s sign, a sensation of electric shock extending down the spine and/or extremities triggered by the flexion of the neck, is a paroxysmal MS-induced neuropathic pain syndrome and is mediated by demyelinating plaques in the cervical spine that activate ascending spinothalamic tracts [24,42]. Motor function impairment presents almost invariably in patients with MS during the course of the disease and consists mostly in pyramidal signs: paresis, spasticity, and hyperreflexia. The demyelinating process can also localize in the cerebellum and the brainstem, causing symptoms such as ataxia, nystagmus, and gait alterations [43]. Optic neuritis usually affects the anterior segment of the optic nerve and clinically manifests as a sudden onset of ocular pain, visual loss with a blind spot in the centre of the visual field, and coloured vision dysfunction (dyschromatopsia). Nearly one-fifth of patients with MS experience optic neuritis as presenting manifestation; among subjects with optic neuritis, 35–70% develop MS over 10 years [44]. Sexual and sphincter dysfunction occurs in more than half of patients and usually follows the involvement of the autonomic nervous system [45]. 

Classically, four clinical courses of MS have been distinguished: relapsing–remitting, secondary progressive, primary progressive, and progressive relapsing. In most patients, the onset of MS consists of an unpredictable and sudden occurrence of neurologic dysfunction, an event named clinically isolated syndrome [24]. A longitudinal study evaluated the natural clinical history of patients with clinically isolated syndrome: at 20 years after onset, 44.7% of patients developed relapsing–remitting MS, 38.8% the secondary progressive form, whereas 12.8% were still clinically isolated syndrome [47,48]. The relapsing–remitting form is the most frequent (72% of patients), followed by the secondary progressive (12%) and the primary progressive (15%) [48,49]. In the relapsing–remitting form, subsequent demyelinating flares are followed by complete resolution, while in the secondary progressive disease, after an initial relapsing course, resolution becomes only partial and damage progressively accumulates. In the primary progressive subset, the accumulation of damage starts from disease onset. Although some differences in laboratory markers and imaging findings between these MS subtypes have been described, there is not enough evidence to formulate a serological or imaging-based classification; thus, the actual subtypes are solely based on the clinical course of the disease [50]. 

Magnetic resonance imaging (MRI) is regarded as the most useful imaging test, with both diagnostic and prognostic implications as it allows for identifying both active and older demyelinating plaques [9,36]. Characteristic lesions present ovoidal shape and are located in specific sites, most commonly in the white matter close to the ventricles (periventricular lesions) or to the cortex (juxtacortical lesions), and in infratentorial areas. The T1-weighted MRI study with gadolinium-based contrast injection is the cornerstone in the diagnostic approach to patients with MS, allowing one to distinguish recent and active demyelinating lesions (enhancing after contrast agent injection) from dated non-enhancing lesions [51]. Lesions should be present in at least two CNS areas (dissemination in space); in addition, non-enhancing and enhancing lesions should be concomitantly present at any time or new lesions should supervene with reference to a baseline MRI scan (dissemination in time) [36]. According to 2017 revision of McDonald’s classification criteria, dissemination in time can be replaced by the evidence of oligoclonal bands in cerebrospinal fluids. MS patients often display IgG oligoclonal bands in cerebrospinal fluid that are typically absent in the serum together with an elevated IgG index. Such an update is of importance, as it allows one to formulate a MS diagnosis even after the first demyelinating attack [36,45].

### 1.4. Beyond the Central Nervous System: Uveitis and Multiple Sclerosis

Although not common, uveitis can present in the context of MS. Based on the largest retrospective studies considering more than 1000 subjects, the prevalence of MS among uveitis patients can be estimated at 1.03% (0.9–1.7%). Conversely, mean prevalence of uveitis among patients with MS is reported to be 0.82% (0.65% to 1.1%). Both figures are significantly higher than those reported in the general population, thus strengthening the relationship between MS and uveitis [52]. Among MS patients, intermediate uveitis is the most common anatomical diagnosis (45–94%), followed by panuveitis (33–50%); anterior uveitis, when present, is typically granulomatous (12.5–100%) [53,54,55,56,57].

Among patients with intermediate uveitis, instead, MS is the second most common aetiology after the idiopathic form named pars planitis, an inflammation of pars plana, the narrowed area between the iris and the choroid. Clinically, pars planitis may manifest as blurred vision or dark, floating spots in the vision and progressive vision loss [58]. Cystoid macular oedema, epiretinal membrane, and cataract are relatively common complications, and potentially sight threatening [59,60,61]. However, the visual outcome of uveitis associated with MS is usually favourable, provided that ocular inflammation is managed properly [62].

Many uncertainties relate to the pathophysiology, given the rarity of the disorder and patient heterogeneity. The etiological relationship between MS and uveitis may be partially explained by the common embryological origin between ocular and neural tissues. Moreover, MS and intermediate uveitis are both associated with HLA-DRB1*15:01. T and B lymphocytes may have a crucial role in both diseases, but further research is needed to establish the exact pathogenic mechanism [63].

### 1.5. Multiple Sclerosis and Other Autoimmune Diseases: Deciphering the Impact of Comorbidities

The association between MS and other autoimmune diseases is controversial: several studies have analysed the incidence and the prevalence of autoimmune conditions in MS patients, but results are not univocal. In 2000, a French study showed a significantly higher rate of autoimmune disorders among patients with MS [64], while another study could not demonstrate a statistically higher incidence of autoimmune comorbidities [65]. A 2008 Danish nationwide cohort study found a higher incidence of ulcerative colitis (RR 2.0) and pemphigoid (RR 15.4), but a lower incidence of temporal arteritis (RR 0.5) [66]. In 2015, a meta-analysis based on 61 studies (more than a half of them were of poor quality) reported psoriasis and autoimmune thyroiditis as the two most common autoimmune comorbidities in patients with MS (7.74% and 6.44%, respectively) [67]. Focusing on the rheumatologic field, almost all studies did not describe a higher risk for patients with MS of developing spondyloarthritis (SpA) [68,69,70] or RA [69,70,71,72], whereas a single study reports a lower RA rate in the same population (RR 0.5) [66]. Given these conflicting data, a meta-analysis did not demonstrate a higher risk of developing RA in patients with MS [73]. It has been suggested that citrullinated CNS proteins, including glial fibrillary acidic protein and myelin basic protein, may contribute to MS pathogenesis by triggering autoimmune mechanisms. Citrullination is a post-translational modification resulting in the conversion of arginine to citrulline, catalysed by the peptidyl arginine deaminase enzyme [74,75,76]. Anti-citrullinated peptide antibodies (ACPA) are the serum biomarkers of RA and polyarticular JIA [77,78]. ACPA positivity could not be detected in any of the 38 MS patients enrolled in a recent case-control study [79].

### 1.6. The Crossroad between Rheumatological Conditions and Multiple Sclerosis: Agents Targeting TNF-α

Given the above discussed relevance of TNF in the pathogenesis of MS, in the early years, TNF inhibitors were proposed as potentially useful tools in the management of MS. Highly unexpectedly, few anecdotic reports described how the neurologic status of MS patients undergoing anti-TNF-α treatment significantly worsened [36]. Most importantly, the first MS clinical trial with a nonselective TNF inhibitor named lenercept was early interrupted, as an exacerbation of symptoms was registered in treated patients [80]. However, the detrimental relationship between TNF inhibition and MS goes well beyond the worsening of neurological function of patients diagnosed with MS who received anti-TNF-α agents. Drugs targeting TNF were shown to potentially trigger MS in genetically predisposed individuals [36]. A large register-based study conducted in Northern Europe including patients with RA, SpA and psoriatic arthritis (PsA) reported a significantly higher risk of neuroinflammatory disorders in patients with AS and PsA receiving TNF inhibitors (HR 3.4, 95% CI 1.30–8.96). In the same study, patients with RA did not show a higher risk of neuroinflammation in both the Danish and the Swedish cohorts (0.97, 95% CI 0.72–1.33 and 1.45, 95% CI 0.74–2.81, respectively) [81]. Conversely, a nested case-control study showed how exposure to anti-TNF-α agents significantly increases the risk of developing brain inflammatory lesions, both demyelinating and nondemyelinating, in a cohort of 212 patients with several autoimmune diseases such as RA, SpA, PsA, psoriasis, Crohn’s disease and ulcerative colitis, conveying an adjusted odds ratio (OR) of 3.01. Notably, such association was predominantly observed among RA patients; in such a subpopulation, TNF inhibition conferred an OR as high as 4.82 [82]. 

Although the relationship between treatment with TNF inhibitors and MS has been established, it is currently unclear whether these pharmacological agents trigger the demyelinating disease or rather exacerbate an asymptomatic underlying neurologic inflammation. It has been proposed that the deleterious role of TNF inhibitors in favouring MS progression and onset might be ascribed to the pleiotropic effects of this cytokine. TNF modulates the immune system via two receptors: TNFR1 is expressed by almost all the immune cells, while TNFR2 is found exclusively in T regulatory cells (Treg) [83,84]. Although TNF is widely considered as a proinflammatory cytokine, the discovery of its binding to TNFR2 opened new frontiers in the elucidation of its biological functions. TNFR2 activation results in the promotion of Treg activity, turning down the inflammatory response [83,84]. Such a novel concept of the dual role of TNF could partially explain the paradoxical, although rare, negative effect of TNF inhibitors in some patients with autoimmune diseases [83,84].

An extensive literature search retrieved no reports describing MS onset in patients receiving any other conventional synthetic (methotrexate, leflunomide, sulphasalazine), biological (tocilizumab, abatacept, ustekinumab, secukinumab, ixekizumab) or targeted synthetic DMARDs (baricitinib, tofacitinib, filgotinib, upadacitinib).

## 2. Description of Two Cases

### 2.1. Case 1

A 22-year-old woman had received a diagnosis of oligoarticular JIA at the age of 18 months; one year later, JIA-associated uveitis was diagnosed. She was effectively treated with methotrexate since the age of 10 years, and she was regularly followed at the outpatient clinic of transitional age rheumatology. Anterior bilateral uveitis was still active and complicated by cataract, posterior synechiae, and band keratopathy in both eyes. Thus, she was started on TNF-α inhibitors; infliximab was soon stopped after an infusive adverse reaction. At the age of 20, she was started on adalimumab in association with methotrexate with optimal response. Few attempts to taper adalimumab were taken, but relapses occurred. After two years of such combo treatment, she complained of new-onset dizziness. Vestibular causes of dizziness were excluded and brain CT scan was performed without any pathological finding. However, a brain MRI led to the identification of subcortical and periventricular hyperintense lesions in T2-weighted MRI sequences. Thus, our patient was admitted to the neurology department, and an MRI contrast scan was performed, including both the brain and the spinal cord. Non-enhancing lesions were found in periventricular and supratenturial areas and the dorsal spinal cord. No signs of optic neuritis were described. Characteristic oligoclonal bands were detected in CSF analysis. According to the 2017 McDonald’s criteria, the diagnosis of MS was made. Adalimumab was discontinued, and IV methylprednisolone was administered (1000 mg per day for 3 days); treatment with methotrexate was continued.

After 6 months, brain MRI showed three new active demyelinating lesions despite no new neurological signs or symptoms; spinal cord MRI did not evince any new lesion. After nearly 8 months after adalimumab discontinuation, at ophthalmologic screening, active uveitis was described. Thus, a multidisciplinary evaluation was performed, and treatment with anti-CD20 biological agent (rituximab) was started.

### 2.2. Case 2

A 21-year-old woman received a diagnosis of oligoarticular-extended ANA positive JIA at the age of 18 months; at the age of 3 years, a bilateral anterior uveitis was detected at routine screening, later complicated by band keratopathy. Since the age of nine, she has been treated with methotrexate plus adalimumab for 4 years, and then, methotrexate was stopped for gastric intolerance and replaced by leflunomide. She continued with such association treatment for 4 years when, for a scarce control of joint disease, she was started on infliximab. After one year, infliximab was stopped due to the persistence of arthritis. Abatacept was started and continued for approximately three years, when she became pregnant. She opted for an elective termination of pregnancy, and after 6 months, she complained of unilateral retroauricular and upper limb paraesthesias. The brain MRI with contrast detected non-enhancing demyelinating lesions in supratentorial and subtentorial areas. No signs of optic neuritis were described, and the MRI of the spinal cord could not detect any lesion. The patient was admitted to the neurology department, and 3-days IV methylprednisolone (1000 mg per day) was administered. 

After nearly 9 months from MS onset, uveitis was still inactive while arthritis flare occurred. Abatacept was reintroduced with optimal response. New demyelinating lesions were not found on follow-up brain and spinal cord MRI. 

## 3. When Juvenile Idiopathic Arthritis and Multiple Sclerosis Coexist: The Description of Five Patients in the Literature

MS and JIA share several features as autoimmune pathophysiology [85,86] and a good response to immunosuppressant and biological therapies [87,88]. The coexistence of MS and JIA has been rarely reported, and it has not been systematically evaluated. In the literature, only few cases have been described, exclusively as anecdotal reports [89,90,91,92,93] (Table 3). As a whole, available evidence is limited to five patients with coexisting JIA and MS. Of these subjects, only one patient had received a TNF inhibitor prior to MS onset [92]. Three other JIA patients developed demyelinating lesions after being treated with traditional synthetic DMARDs: methotrexate in two cases, leflunomide in one case [89,91,93]. In a single patient, the onset of MS occurred in the paediatric age: Coskun et al. described the case of a 10-year-old girl with previous epilepsy and cognitive impairment due to cerebral palsy, who was diagnosed with JIA and later developed demyelinating attack after two doses of methotrexate. At this regard, it should be highlighted that MS is rare at paediatric age, and the diagnosis should be carefully evaluated [89]. In paediatric patients, acute disseminated encephalomyelitis (ADEM) provides the most common demyelinating condition and represents a diagnostic challenge for paediatric neurologists. ADEM consists in a widespread inflammatory demyelinating insult affecting the brain and the spinal cord, usually with a monophasic pattern. ADEM lesions typically localize in the white matter, usually following viral or bacterial infections or, less often, vaccination for measles, mumps, or rubella. The onset of ADEM is abrupt, with encephalopathy and multifocal neurological deficits. The usual progression from onset to maximal severity of symptoms occurs over 4–7 days. Common exam findings include altered mental status, ataxia, and extremity weakness. ADEM can also be associated with optic neuritis and transverse myelitis, leading to vision changes and extremity weakness/urinary retention, respectively [94]. Unfortunately, the description of a fifth case of coexistence of JIA and MS in a 31-year-old woman was available only in the Polish language; thus, no relevant data could be extracted [90]. 

When Juvenile Idiopathic Arthritis and Multiple Sclerosis Coexist: A Therapeutic Challenge for Both Rheumatologists and Neurologists 

Although rarely reported, the coexistence of MS and JIA in a single patient poses several challenges in terms of therapeutic management to both rheumatologists and neurologists. The treatment of MS aims not only at handling acute demyelinating flares but even at reducing the flare rate. If intravenous corticosteroids represent the main treatment choice for acute flares, several disease-modifying drugs are currently available for reducing the risk of new demyelinating attacks. The 2018 European Academy of Neurology’s guidelines for the management of MS recommends starting the treatment of patients with the relapsing remitting form, the most common subset, with one among the options listed in Table 4. The therapeutic decision should be tailored upon each patient’s clinical features, eventual comorbidities, and drug availability [95]. Currently, ocrelizumab is the only pharmacological tool that has been formally approved for the treatment of primary progressive and secondary progressive MS; thus, it should be the preferred option in these patients [96]. 

To adequately manage both JIA and MS, neurologists and rheumatologists should find a common therapeutic strategy that should be effective for both conditions. Unfortunately, only few drugs allow one to adequately manage both CNS and joint inflammation: teriflunomide/leflunomide, methotrexate and anti-CD20 agents, according to the severity of the conditions [106].

Teriflunomide is an immunosuppressive drug approved for the treatment of mild to moderate relapsing–remitting MS, while its precursor leflunomide is a conventional synthetic DMARD routinely used in the management of the most common inflammatory arthritides, including JIA [99]. Leflunomide and teriflunomide share the same mechanism of action, which envisages the interference with the de novo pyrimidine biosynthesis via the inhibition of the mitochondrial enzyme dihydroorotate dehydrogenase. Unfortunately, the efficacy of leflunomide in MS has never been tested in patients with MS, nor has teriflunomide in the setting of inflammatory arthritides. Nevertheless, teriflunomide is regarded as a reasonable option in case of overlap of mild/moderate relapsing–remitting MS and inflammatory arthritis [106].

Methotrexate plays a key role in the management of JIA: this folic acid analogue exerts an immunosuppressant and anti-inflammatory action via the inhibition of purine and pyrimidine synthesis [107]. Unfortunately, methotrexate is poorly effective in patients with MS, even though it led to a reduction of the flare rate [108]. Thus, it could be taken into account as a therapeutic option in patients with mild demyelinating disease and concomitant JIA. 

If teriflunomide/leflunomide and methotrexate represent valid choices for the management of mild to moderate MS and concomitant inflammatory arthritis, anti-CD20 monoclonal antibodies are the main option in case of high disease activity. CD20 is a surface protein expressed by B-lymphocytes [109], targeted by anti-CD20 monoclonal antibodies such as rituximab and ocrelizumab. Rituximab is a monoclonal chimeric anti-CD20 antibody, the first drug of this class to be approved for the treatment of RA back in 2006, even if it is now used even in other autoimmune and hematologic conditions [110]. Ocrelizumab is a recently developed anti-CD20 humanized antibody, the only drug demonstrating significant improvement in primary progressive MS [96]. Even though it has not been officially approved, rituximab is frequently prescribed as an off-label drug by neurologists in both relapsing–remitting and progressive MS [110]. However, ocrelizumab is not approved for the treatment of inflammatory arthritides even if it led to a reduction of disease activity and slowed radiographic progression in clinical trials [111,112]. Therefore, anti-CD20 antibodies represent a pivotal tool for rheumatologists and neurologists in the case of concomitant demyelinating lesions and inflammatory arthritis.

Unfortunately, other immunosuppressant drugs routinely exploited in the management of RA and JIA, such as anti-IL6 (tocilizumab, sarilumab) and anti-IL17 agents, lack of sufficient evidence to be integrated into the pharmacological armamentarium of MS; conversely, IL-1 antagonists and abatacept were trialled in MS with poor results [106].

Lately, four Janus Kinase inhibitors, baricitinib, tofacitinib, filgotinib and upadacitinib, have been marketed for the treatment of RA, but only tofacitinib can be prescribed to patients with JIA. JAK pathways are involved in MS as well, and these drugs will probably be evaluated in prospective trials even for demyelinating diseases [113,114].

## 4. Discussion

To the best of our knowledge, the present description of two patients with concomitant JIA and MS is the first in the literature to report the potential interplay between JIA, uveitis and MS. The concomitance of uveitis might be of particular interest, since it might unveil the pathogenic mechanisms involved in the onset of MS. It can be postulated that an underlying inflammatory condition in the ocular structures might support the onset of the characteristic demyelinating lesions in the CNS, given the common embryological origin and the anatomic contiguity. 

Although MS has been frequently associated with other autoimmune diseases, the coexistence of MS and JIA in the same patient is poorly described: previous evidence is limited to five JIA patients without a diagnosis of uveitis who developed demyelinating CNS lesions during follow-up. In our experience, both patients had been exposed to anti-TNF-α agents prior to the onset of demyelinating lesions: one subject had been treated with adalimumab for 2 years at the time of the first demyelinating attack, while the other one was receiving abatacept when neurological symptoms occurred but had previously received TNF-α inhibitors for nearly 8 years (adalimumab and infliximab). The exposure to anti-TNF-α in our patients differs from the cases previously described in the literature: a single patient out of the five reports had been exposed to anti-TNF-α before the onset of MS [91]. The detrimental relationship between treatment with TNF inhibitors and demyelinating diseases has been extensively analysed, and a large burden of evidence supports the association, even though exposure to anti-TNF-α seems to carry a not so burdensome hazard of MS. Several studies focused on MS occurrence in patients treated with TNF-α inhibitors and concluded that these patients have a slightly higher risk of developing MS and demyelinating events. In 69% of cases, MS onset presents within 5 years after starting TNF-α agents and in 37% after 2 years [115,116]. First, In the specific settings of JIA, it still remains to be clarified whether there is an increased rate of demyelinating disease. Second, it should be elucidated if MS and JIA represent comorbid conditions, or rather the potential augmented hazard of MS is underpinned by anti-TNF-α treatment. Undoubtedly, as already discussed, TNF-α is not only a key cytokine in the pathophysiology of almost all inflammatory arthritides but even a top pathogenic player in the onset and progression of MS demyelinating process. 

As in almost all the previously described cases, our patients had been exposed to methotrexate before neurological symptoms occurred. However, methotrexate treatment has never been associated with an increased risk of MS: methotrexate is not contraindicated in patients developing demyelinating lesions, and it has even demonstrated a beneficial effect in MS management [108].

The coexistence of JIA and MS in the same subject requires a close multidisciplinary collaboration; neurologists and rheumatologists face several difficulties when optimizing the therapeutic approach to patients with these two conditions. Currently, available evidence supports teriflunomide and anti-CD20 agents as the two best tools since they are active on both sides of inflammations, the CNS as well as the joints. The choice should be based on the severity of both MS and JIA, and they adequately account for the response to previous treatments and potential extra-articular manifestations and comorbidities. 

Two of the patients described in the literature had neurological disease treated with IFN-γ, and a good clinical response was registered in both instances [91,92]. Although IFN-γ was the earliest disease-modifying drug approved for the management of MS, it still represents an important first-line therapeutic option despite a poor safety profile [117]. IFN-γ has been demonstrated to have anti-inflammatory properties in synovial tissue reducing the production of cytokines involved in the pathogenesis of inflammatory arthritides, such as TNF-α and IL-6. For this reason, the potential use of IFN-γ in patients with RA and other inflammatory arthritis had been suggested [118], and MS patients treated with IFN-γ could experience relief of joint inflammation as well as neurological involvement. 

As a whole, our experience of two young women with JIA presenting with MS strongly suggests the pivotal importance of maintaining strict clinical vigilance over the onset of novel neurological signs and/or symptoms, requesting the appropriate imaging investigations and timely referring to the neurologist. Given the strong interplay between MS onset and anti-TNF-α agents emerged in particular in the setting of RA, biological agents targeting TNF-α should be promptly discontinued, even if JIA-related data are still lacking. In addition, these pharmacological compounds should be avoided in patients at high risk for MS or who have a history of demyelinating events. Large, multi-centre, prospective studies are highly warranted in order to clarify whether for JIA patients MS represents a comorbidity or rather a treatment-related complication.

## Figures and Tables

**Figure 1 biomedicines-10-02041-f001:**
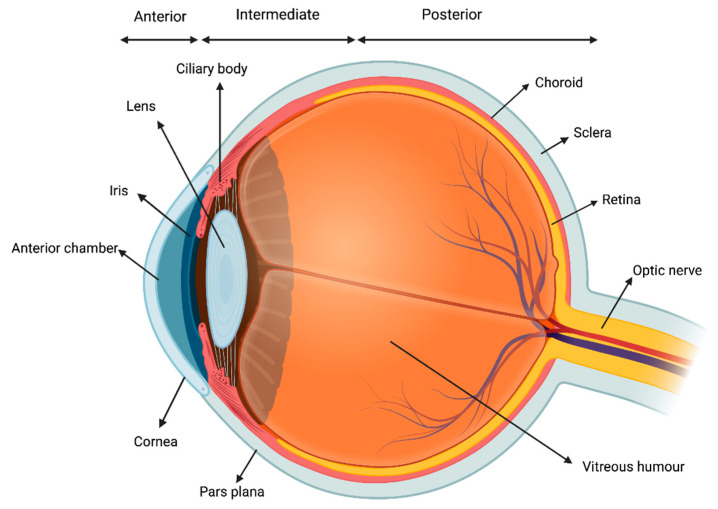
Visual representation of the uvea.

**Table 1 biomedicines-10-02041-t001:** The 2017 revision of McDonald’s diagnostic criteria for multiple sclerosis [41].

Dissemination in Space (DIS)	Dissemination in Time (DIT)
One or more lesions in two or more of the following sites: -Periventricular -Cortical/Iuxtacortical -Infratentorial -Spinal cord	-New T2 and/or contrast enhancing lesion on subsequent MRI with respect to a baseline scan, independently from the timing of the baseline MRI In patients fulfilling DIS criteria but not DIT, in case of the presence of oligoclonal band in cerebrospinal fluid, MS may be diagnosed

**Table 2 biomedicines-10-02041-t002:** Most common clinical manifestations of multiple sclerosis at any time during disease course.

Weakness	89%
Sensory disturbances	87%
Gait disturbance	82%
Bladder problems	71%
Fatigue	57%
Cramps	52%
Diplopia	51%
Other visual disturbances	49%
Bowel disturbances	42%
Dysarthria	37%
Vertigo	36%
Facial pain	35%

Adapted from Richards RG et al. A review of the natural history and epidemiology of multiple sclerosis: implications for resource allocation and health economic models. Health Technol Assess 2002 [46].

**Table 3 biomedicines-10-02041-t003:** Literature findings on concurrence of MS and JIA.

First Author, Year	Patients	JIA Subset, Disease Duration	Previous Treatment (s)	Ongoing JIA Treatment	MS Subset	MS Treatment
Ozsahin et al., 2014 [91]	1 patient (10 y, F)	Oligoarticular JIA, 1 year	Naproxene 250 mg/day	MTX (two weeks)	ADEM	Not specified
Coskun et al., 2011 [89]	1 patient (35 y, F)	JIA, 29 years	MTX, 5-ASA, intermittent CCS	None (poor compliance)	Not specified	CCS pulses → IFNβ
Sicotte et al., 2001 [92]	1 patient (21 y, F)	Polyarticular JIA, 13 years	Oral and i.m. gold, MTX, SSZ	ETN (9 months), celecoxib	Not specified	CCS pulses → leflunomide → IFNβ
Kaouther et al., 2011 [93]	1 patient (21 y, F)	Polyarticular JIA, 5 years	None	LEF 20 mg/day	Not specified	CCS pulses

5-ASA: 5-aminosalicylic acid (mesalazine); ADEM: acute diffuse encephalomyelopathy; CCS: corticosteroids; ETN: etanercept; IFN-β: interferon-β; JIA: juvenile idiopathic arthritis; LEF: leflunomide; MS: multiple sclerosis; MTX: methotrexate; SSZ: sulfasalazine.

**Table 4 biomedicines-10-02041-t004:** Mechanisms of action of first line drugs for multiple sclerosis.

Drugs	Mechanism of Action
Interferon-β	Reduces the trafficking of inflammatory cells across the blood-brain barrier Increases of nerve growth factor production Increases the number of CD56+ natural killer cells in the peripheral blood
Glatiramer acetate [97]	Modulates the activity of antigen presenting cells Probably inhibits the activity of B cells
Dimethyl fumarate [98]	Anti-inflammatory action via type II myeloid cells and T helper 2 cells
Teriflunomide [99]	Reduces proliferation of B and T cells blocking the de novo pyrimidine synthesis pathways
Fingolimod [100]	Inhibits the egression of lymphocytes from lymphoid tissues
Cladribine [101]	Reduces T and B circulating cells interfering with enzymes involved in DNA metabolism
Daclizumab [102]	Expands CD56+ natural killer cells Blocks the cross presentation of IL-2 by dendritic cells to T helper cells
Natalizumab [103]	Inhibits the migration of inflammatory cells across the blood–brain barrier blocking the integrin very late antigen-4 (VLA-4)
Ocrelizumab [104]	Suppresses immune response blocking CD20+ B cells
Alemtuzumab [105]	Determines long-lasting depletion of CD52+ T and B cells

## Data Availability

Not applicable.

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
