# Peer review of "Juvenile Idiopathic Arthritis, Uveitis and Multiple Sclerosis: Description of Two Patients and Literature Review"

_biomedicines, 2022, doi:10.3390/biomedicines10082041_

Round 1

Reviewer 1 Report

Authors raise two unusual cases JIA concominant with newly Dx of MS and review literatures. First importnat concern is that did you check AQP4 Ab and/or MOG Ab to exclude NMOSD or MOG-encephalopathy ?

Whether two cases have optic neuritis, please make a clear mention. 

Second, Please cite the brain and spinal cord MRI to support your description.

Line 503 and 504, "JIA and SM" is it correct? Please check it.

Author Response

We thank the referee for the positive evaluation of our manuscript.

Please find below a response to the criticisms raised:

1) First importnat concern is that did you check AQP4 Ab and/or MOG Ab to exclude NMOSD or MOG-encephalopathy ?

Whether two cases have optic neuritis, please make a clear mention. 

None of the patients had optic neuritis, as now clearly stated in the text.

Therefore, there was no clinical indication to test antibodies against AQP4 or to exclude MOG-encephalopathy.

2) Second, Please cite the brain and spinal cord MRI to support your description.

Details on spinal cord MRI have been added in the text.

3) Line 503 and 504, "JIA and SM" is it correct? Please check it.

We thank the referee for noticing this typo, it has been edited.

Reviewer 2 Report

I completed the revision of the manuscript entitled  “Juvenile idiopathic arthritis, uveitis and multiple sclerosis: 2 description of two patients and literature review”.  The manuscript by Cecilia Beatrice Chighizola et al. is very interesting. The structure of the manuscript (Introduction, Case Report, Discussion) follows a logical sequence. The introduction contains sufficient background information.  The next section comprehensively describes two cases of patients with JIA and MS and the treatment of these diseases. The Discussion provides a good summary.

Specific minor comments:

1.       The ciliary body should be signed in Fig. 1.

2.       Line 150-151 should be: Several risk factors for JIA have been identified: ANA positivity, younger age (<6 years of age) at the onset of arthritis, oligoarticular subset, and female gender are the main ones.

3.       line 405 and 437: wrong numbering of point 3 followed by point 2.4 - should be systematized

4.       The references is presented carelessly. In one item, the authors give the abbreviations of the journals, and in another the full names

Author Response

We thank the referee for the positive evaluation of our manuscript.

Please find below a response to the criticisms raised:

1)     The ciliary body should be signed in Fig. 1.

Figure 1 has been edited by adding the ciliary body sign.

2)      Line 150-151 should be: Several risk factors for JIA have been identified: ANA positivity, younger age (<6 years of age) at the onset of arthritis, oligoarticular subset, and female gender are the main ones.

The statement has been edited has suggested by the referee.

3)       line 405 and 437: wrong numbering of point 3 followed by point 2.4 - should be systematized.

The numbering has been systematized.

4)      The references is presented carelessly. In one item, the authors give the abbreviations of the journals, and in another the full names

The reference list has been edited.

Reviewer 3 Report

Thank you for submitting this paper, it was overall very interisting to read. 

I have a few suggestions that need to be adressed:

A lot more citations are needed:

Could you provide a source for the claim:

1. made in the introduction, that antiTNFalpha treatment in RA is associated to MS.

2. about sJIA extra-articular manifestations.

3. Celltypes and cytokines involved in sJIA.

4. Treatment strategies for JIA overall (Treat-to-target studies)

5. T cell involvement in uveitis.

6. clinical courses of MS.

7. Magnetic resonance imaging (MRI) is regarded as the most useful imaging test, with 258 both diagnostic and prognostic implications as it allows identifying both active and older 259 demyelinating plaques.

8. Characteristic lesions present ovoidal shape and are located in 260 specific sites, most commonly in the white matter close to the ventricles (periventricular 261 lesions) or to the cortex (juxtacortical lesions), and in infratentorial areas.

9. Lesions should be present in at least two CNS areas (dissemination in space); in addition, 266 non-enhancing and enhancing lesions should be concomitantly present at any time or new 267 lesions should supervene with reference to a baseline MRI scan (dissemination in time).

10. 

Please rephrase Table 1 it looks like presence of oligoclonal band in CSF ist both dissemination in space (wording) and in time (location in table).As far as I understand the oligoclonal band represents DIT if DIS criteria are met. Also check for proper easy readability by better arragning the table. 

11. while other studies could not 302 demonstrate a statistically higher incidence of autoimmune comorbidities [60]. -> other studies implies more than 1 study.

12. Highly unexpectedly, few anecdotic reports described how the neurologic status of MS 324 patients undergoing anti-TNF- treatment significantly worsened.

13. Indeed, TNFR2 activation results in the promotion of Treg activity, turning down the 352 inflammatory response.

14. 

Given the above discussed relevance of TNF in the pathogenesis of MS, in the early 322 years TNF inhibitors were proposed as potentially useful tools in the management of MS. -> the relevance of TNF was not discussed in the section above. please aelaborate more on TNFalpha. 

Table 4 is not well formattted which drug does which mechanism belong to?

Overall I would suggest to shorten the introduction, put more details into the discussion of the overall now 7 cases.

some minor spellchecks needed (SM instead of MS, simptoms...)

Author Response

We thank the referee for the positive evaluation of our manuscript.

Please find below a response to the criticisms raised:

1) A lot more citations are needed.

Citations of references already included in the reference list have been added in the text, as suggested by the referee. New references have also been cited.

  1. made in the introduction, that antiTNFalpha treatment in RA is associated to MS.
  2. about sJIA extra-articular manifestations.
  3. Celltypes and cytokines involved in sJIA.
  4. Treatment strategies for JIA overall (Treat-to-target studies)
  5. T cell involvement in uveitis.
  6. clinical courses of MS.
  7. Magnetic resonance imaging (MRI) is regarded as the most useful imaging test, with 258 both diagnostic and prognostic implications as it allows identifying both active and older 259 demyelinating plaques.
  8. Characteristic lesions present ovoidal shape and are located in 260 specific sites, most commonly in the white matter close to the ventricles (periventricular 261 lesions) or to the cortex (juxtacortical lesions), and in infratentorial areas.
  9. Lesions should be present in at least two CNS areas (dissemination in space); in addition, 266 non-enhancing and enhancing lesions should be concomitantly present at any time or new 267 lesions should supervene with reference to a baseline MRI scan (dissemination in time).
  10. Please rephrase Table 1 it looks like presence of oligoclonal band in CSF ist both dissemination in space (wording) and in time (location in table).As far as I understand the oligoclonal band represents DIT if DIS criteria are met. Also check for proper easy readability by better arragning the table. 

Table 1 has been edited as suggested by the referee.

  1. while other studies could not 302 demonstrate a statistically higher incidence of autoimmune comorbidities [60]. -> other studies implies more than 1 study.

The text has been edited to: Another study....

  1. Highly unexpectedly, few anecdotic reports described how the neurologic status of MS 324 patients undergoing anti-TNF- treatment significantly worsened.
  2. Indeed, TNFR2 activation results in the promotion of Treg activity, turning down the 352 inflammatory response.
  3.  

Given the above discussed relevance of TNF in the pathogenesis of MS, in the early 322 years TNF inhibitors were proposed as potentially useful tools in the management of MS. -> the relevance of TNF was not discussed in the section above. please aelaborate more on TNFalpha. 

The dissertion of the role of TNF in the pathogenesis of MS has been expanded, as suggested by the referee.

- Table 4 is not well formattted which drug does which mechanism belong to?

Table 4 has been reformatted.

- Overall I would suggest to shorten the introduction, put more details into the discussion of the overall now 7 cases.

We thank the referee for the suggestion, however we believe that presenting our two cases separately allows to focus on the peculiarities of their clinical presentation. In the Discussion section, we point out the dissimilarities and commonalities between our cases and the 5 already described in literature.

We hope that our choice is acceptable for the referee.

- some minor spellchecks needed (SM instead of MS, simptoms...)

The typos have been edited, thank you for noticing.